# Exploring How LLMs Capture and Represent Domain-Specific Knowledge

## Abstract

We study whether Large Language Models (LLMs) inherently capture domain-specific nuances in natural language. Our experiments probe the domain sensitivity of LLMs by examining their ability to distinguish queries from different domains using hidden states generated during the prefill phase. We reveal *latent domain-related trajectories* that indicate the model's internal recognition of query domains. We also study the robustness of these domain representations to variations in prompt styles and sources. Our approach leverages these representations for model selection, mapping the LLM that best matches the domain trace of the input query (i.e., the model with the highest performance on similar traces). Our findings show that LLMs can differentiate queries for *related domains*, and that the fine-tuned model is not always the most accurate. Unlike previous work, our interpretations apply to both closed and open-ended generative tasks.

## 1 Introduction

Large Language Models (LLMs) have demonstrated remarkable capabilities across various tasks, yet the internal mechanisms driving these capabilities remain poorly understood. Different domains require distinct knowledge and reasoning patterns, necessitating LLMs to adjust decision-making based on-the-fly for input queries. This is crucial for applications demanding high reliability, such as legal and medical fields, where errors can lead to significant consequences.

The research question of *how LLMs adapt their decision-making and reasoning patterns across different domains* is distinct from a growing body of work on locating factual associations from language models behavior (Meng et al., 2024; Hernandez et al., 2024a;b; Mitchell et al., 2022; Meng et al., 2023; Dai et al., 2022; Belrose et al., 2023). While these studies aim to identify the modules and computations that recall specific facts, primarily monitoring and controlling language generation, they often fall short in addressing the complexities of generative tasks.

Understanding how LLMs adapt their reasoning across generative tasks is important for enhancing transparency in their decision-making processes. This insight not only deepens our understanding of generalization capabilities but also promotes interdisciplinary collaboration and improves the design of evaluation metrics that consider domain-specific nuances. Our research focuses on the patterns models reveal as they tackle domain-specific challenges, rather than merely retrieving factual information.

Recently, Guo et al. (2024) evaluated GPT-4's ability to infer domain knowledge using a ReAct-based LLM chain. The experiment involved generating reasoning paths and actions from unlabeled coding exemplars without explicit domain descriptions. Their findings show that GPT-4, when given domain-relevant exemplars, significantly outperforms its generic counterpart, suggesting that the model can discern domain essence from the exemplars. However, it remains unclear whether the model truly "understands" the content or merely imitates the exemplars based on its outputs. Similarly, other efforts focus on creating probing representations for individual context-dependent situations Li et al. (2021); Pimentel et al. (2020), where performance varies significantly based on task-specific metrics.

Our research, motivated by studies on neural network activation Abdelnabi et al. (2024); He et al. (2024); Mallen & Belrose (2024), aims to interpret how hidden states represent context for domain-related queries before the generation phase. We aim to determine if models inherently encode gen-

eral natural language for specific domains. Our work builds on previous research Mallen & Belrose (2024); Burns et al. (2024); He et al. (2024), which focused on probing mechanisms for closed-ended tasks. In contrast, we explore hidden states in open-ended scenarios, offering a clearer understanding of domain nuances across different LLMs.

**Overview of results and main contributions.** Our results show the power of hidden state activations as domain representations. We analyze hidden state traces across multiple LLM architectures – Gemma (Mesnard et al., 2024), Phi (Abdin et al., 2024), Mistral (Jiang et al., 2023a), and Llama (Touvron et al., 2023) – and found consistent patterns in domain-specific activations, even with variations in prompt styles and instructions. This consistency suggests hidden states capture fundamental domain characteristics rather than superficial textual features. Our comparative study with traditional methods, such as semantic routing (Manias et al., 2024b; Labs, 2024) and token-based classification (He et al., 2021), demonstrates the potential advantages of using internal model representations for domain interpretation. Our main contributions are as follows:

- **Latent domain representations:** We demonstrate that hidden states in LLMs capture domain-specific information, which remains robust across multiple architectures and prompt variations. These hidden states activations show consistent separation across domains, providing a powerful signal for identifying the underlying domain of a query. We name these signals *latent domain-related trajectories*.

- **Robustness across tasks and models:** We show that *latent domain-related trajectories* are consistent across various LLM architectures and remain stable even after fine-tuning. This opens up new possibilities for efficient model selection, especially in tasks requiring cross-domain generalization, such as legal, medical and mathematical reasoning.

- **Improved model selection:** Our experiments show that leveraging the *latent domain-related trajectories* for model selection, leads to significant performance improvements compared to traditional semantic and token-based methods. Specifically, the LLM Hidden States Classifier achieves a 12.3% accuracy improvement over domain fine-tuned models, showing particular strength on open-ended tasks like GSM8K Cobbe et al. (2021) and MATH Hendrycks et al. (2021b).

## 2 RELATED WORK

**Understanding Transformers-based Models.** Transformers (Vaswani et al., 2017) play a key role in Natural Language Processing tasks. As a result, understanding their internal working mechanisms is critical. Research on interpreting Transformer states is based on forwarding data into the model to analyze attention heads (Clark et al., 2019; Abnar & Zuidema, 2020) and embedding spaces (Dar et al., 2023; Geva et al., 2022; 2021) that connect "interpretability" with different data distributions and equivalent predictions. However, these techniques are task-specific and not related to gradient-based measures of feature importance (Jain & Wallace, 2019).

**LLMs Hidden States as Internal Representations:** Hidden states have been studied to investigate factual knowledge (He et al., 2024; Chen et al., 2024; Burns et al., 2023), hallucination (Zhao et al., 2024; Dombrowski & Corlouer, 2024), locating and modifying factual data (Meng et al., 2022; Hernandez et al., 2024a) and task drifts (Abdelnabi et al., 2024; Zverev et al., 2024). Most research is limited to closed-ended scenarios that involve probing a white-box model to uncover contrasting behaviors (often impractical for generative tasks). Bricken et al. (2023) decompose activations into more interpretable monosemantic features using a sparse autoencoder. Feature decomposition can determine the contribution of the layers' activations on a specific example, making it easier to monitor the network activation for specific features. In contrast, we aim to reuse the hidden states generated by the LLM from the context, without using an external autoencoder.

**Domain Representations for Routing Mechanisms.** Semantic Layer approaches (Sun et al., 2024; Manias et al., 2024a) have emerged as a particularly lightweight and effective solution: by comparing the embeddings of semantic representations (e.g. cosine, Manhattan distances), these layers perform a preselection of language models or tools that need to be retrieved for specific domain tasks. These methods can be restricted when data is scarce, or we do not have a predefined instruction structure. Within the recommendation systems literature, there are works that leverage deep neural networks discriminatively to learn better representations of users/items based on con-

textual information that can be used for downstream tasks (Liang et al.; Li et al., 2023). Yet, to our knowledge, none of these have delved into how to generalize to more complex generative tasks. Alternative routing strategies for model selection (Ding et al., 2024; Ong et al., 2024; Šakota et al., 2024; Jiang et al., 2023b) aim to estimate query complexity and redirect "easy" requests to smaller LLMs, balancing model performance and inference costs. Within this area, some routers based on domain clustering have emerged (Pichlmeier et al., 2024; Ostapenko et al., 2024), demonstrating the ability to efficiently distribute incoming requests by directing them to the nearest cluster of instructions. A limitation of these approaches is their dependency on access to a subset of the expert/cluster training data, which must be adequately representative for comparison purposes, a requirement that may be infeasible for models trained on proprietary data.

In contrast with previous work, our research aims to investigate whether *LLMs can inherently distinguish between queries from various domains, despite differences in prompt style and source* and show which value these representations provide, compared with other semantic and token-level representations.

## 3  PRELIMINARIES

**Operational definitions:** To assess the ability of LLMs to capture domain-specific representations, we rely on hidden states generated during the *prefill phase* – the stage in which the model processes input tokens to generate intermediate states before producing the first new token. Below are the key operational terms used throughout our experiments:

- **Hidden states** are the intermediate activations produced by the model at each layer when processing input tokens, building the model's contextual understanding. For each query, the model generates a set of hidden states in the shape $(batch\_size, dimension, num\_layers)$, where $batch\_size$ refers to the number of samples processed at once, *dim* refers to the size of the hidden representation (i.e., the number of features in each state) and *num_layers* indicates the total number of layers in the model, each producing its own hidden states.

- **Mean activation**: To simplify analysis, we compute the mean activation across both the batch and dimension axes. The mean for each layer $l$ is given by:

$$\mu_l = \frac{1}{\text{batch\_size} \times \text{dim}} \sum_{b=1}^{\text{batch\_size}} \sum_{d=1}^{\text{dim}} A_{b,d,l} \tag{1}$$

This results in a single vector of activations, capturing the average behavior of each layer during the prefill phase.

- **Variance of activations**: We also compute the variance for each layer to measure the spread of the hidden states across samples and dimensions. The variance for each layer $l$ is computed as:

$$\sigma_l^2 = \frac{1}{\text{batch\_size} \times \text{dim}} \sum_{b=1}^{\text{batch\_size}} \sum_{d=1}^{\text{dim}} (A_{b,d,l} - \mu_l)^2, \tag{2}$$

The variance provides insight into how sensitive different layers are to variations in the input. In some sections, we replace the variance computation with standard deviation by only computing its square root.

- **Latent Domain-Related Trajectories**: These refer to the patterns observed in the hidden states that align with specific domains (e.g., Biomedical, Law, Maths). By analyzing the mean and variance of activations across layers, we can trace the model's internal representation of domain-related information.

Through these operational definitions, we quantify the informativeness of hidden state activations, enabling us to investigate whether these states encode meaningful domain-specific knowledge before the generation phase.

**Motivation and Main hypothesis:** Figure 1 illustrates the key observation that motivated our research: queries belonging to similar domains tend to cluster closely when viewed through the lens of hidden state activations. This clustering occurs for both the mean and variance of activations across

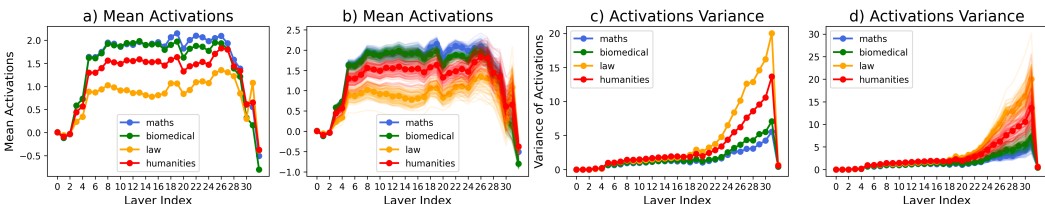

Figure 1: Activation summary produced by Phi-3-mini-3.8B on the MMLU benchmark. The left side shows the mean activation per domain subset (a) and per sample (b), while the right side presents the variance across domains (c) and samples (d).

layers, suggesting that the model exhibits similar "confidence" in processing queries from the same domain.[1]

Building on these observations, we propose the following main hypothesis: *LLMs' hidden states encode generalizable representations for specific domains, revealing domain-related traces from the context understanding phase.* Testing this hypothesis requires:

- **Generalizability:** We aim to determine whether this ability is consistent across different LLM architectures, training recipes, and model parameters. We also investigate whether these representations are retained after fine-tuning and assess how robust they are when prompts are perturbed.

- **Evaluation:** We develop a method that leverages these hidden state representations to benchmark model performance against traditional approaches. Specifically, we quantify the value of these representations compared to token-based and semantic representations.

The experiments and analyses developed in the next sections provide evidence for this hypothesis and evaluate the robustness of hidden states across various models, prompts, and domains.

## 4 EXPERIMENTAL SETUP

We test our hypothesis through controlled experiments analyzing generation and evaluating the performance capabilities of the model across three fields where the accuracy and rationale behind the decision are critical: Healthcare, Finance, and Law. Since our findings are primarily experiment-based, it makes sense to begin by describing the setup and the scenarios we have considered.

**Model Architectures:** We use the DeBERTa (He et al., 2021) encoder model and four different pretrained LLM architectures with public checkpoints available at HuggingFace. Gemma-2B (Mesnard et al., 2024) is an 18-layer LLM with 2B parameters. Phi-3-mini-3.8B (Abdin et al., 2024) is a 32-layer LLM with 3.8B parameters. Llama2-7B (Touvron et al., 2023) and Mistral-7B (Jiang et al., 2023a) are 32-layer LLMs with 7B parameters. We selected these models due to their demonstrated efficacy across a range of tasks and their varying dimensions and training recipes, allowing us to explore the generalization in our findings. We run all experiments on 4 NVIDIA RTX A6000 GPUs with 44 GB of Memory.

**Datasets:** We leverage the multi-domain query nature of the MMLU dataset (Hendrycks et al., 2021a). A total of 30 subtasks were randomly selected from the 57 present in the dataset. Since these subcategories included overlapping domains (e.g. college mathematics, high school mathematics, elementary mathematics can all be categorized within the maths domain), the *supercategories* labels provided by the dataset authors were used to reduce the original 30 subcategories into 4, related to the domain of the question: mathematics, biomedical, law and humanities.[2] This was done to prevent any ambiguity and ensure the results were more comprehensible. For simplicity, in the next sections, we call *Base Pool* (7358 samples) the queries coming from these distributions. We also

---

[1]While this behavior hints at the model's ability to internally distinguish domains, the relationship between the domains is not entirely clear. For example, initial observations show that mathematical and biomedical domains are closely related. While this proximity could raise concerns about the model's ability to fully differentiate between domains, it should be noted that this is a preliminary observation intended to motivate further investigation. Further Analysis on the overlapping of these domains is provided in Appendix A.5

[2]The original subcategories utilized per domain are itemized in Appendix A.1

have included a *Specialized Pool* with a set of domain-specific datasets containing banks of open and closed questions with different types of instructions, covering overlapping domains as MMLU partitions. The GSM8K (Cobbe et al., 2021) dataset probes the informal mathematical reasoning ability. The MEDMCQA Pal et al. (2022) dataset tests the model understanding across a wide range of 21 medical subjects. The CaseHOLD (Zheng et al., 2021) dataset requires identification of legal holdings on cited cases. The Plato dataset contains articles from the Stanford Encyclopedia of Philosophy; the task is to identify different philosophic terms through the passages.

**Baselines and Implementations:** We compare our approach of using LLM hidden states as domain representations with two baselines: a Semantic Layer and a DeBERTa classifier. The **Semantic Layer** (Labs, 2024) performs the model selection based on similarity scores on provided few-shot utterances. We configured this layer with four main routes, each belonging to a domain in the dataset. We specified 1,000 utterances for each route, with queries sampled randomly from MMLU domains, totaling 4,000. [3] We used the default configuration of the HuggingFaceEncoder class for the encoder, which uses the sentence-transformers/all-MiniLM-L6-v2 model with a score threshold of 0.5. We fine-tuned a **DeBERTa** (He et al., 2021) encoder on a sequence classification task. The model was fine-tuned on a classification task with four output labels, each corresponding to a domain (maths, biomedical, law, and humanities). We trained the model on 4,000 samples from the MMLU domains, using a training batch size of 1, a learning rate of 2e-5, and a weight decay of 1e-2 for three epochs. The best model was retained at the end of training. We used a maximum sequence length of 512 tokens and truncated the input sequences to this length. DeBERTa has been chosen as the discriminator due to its remarkable accuracy and performance across several NLP tasks, particularly encoder models. This is supported by its exceptional performance across various model selection frameworks (Ding et al., 2024; Ong et al., 2024; Šakota et al., 2024; Jiang et al., 2023b).

## 5 LATENT DOMAIN-RELATED TRAJECTORIES

### 5.1 ANALYZING THE POWER OF THE LLM HIDDEN STATES

Based on our initial observations on the behavior of the Phi-3-mini-3.8B model, we aim to investigate whether the ability to encode domain-specific information in hidden states is an emergent property of LLMs in general, rather than a model-specific phenomenon. To this end, we conducted a comparative analysis between different generative LLMs (Gemma-2B, Phi-3-mini-3.8B, Llama2-7B and Mistral-7B) and a pretrained encoder model (DeBERTa) to explore the contextual representations captured during the prefill phase. We structured our investigation around the following key questions:

a) **Comparison with encoder models**: How do the hidden states of generative language models compare with those of an encoder model designed to capture more fine-grained semantic and positional information?

b) **Impact of finetuning**: After fine-tuning on specific tasks, do LLMs retain the same domain-specific hidden state traces, or do these traces shift significantly?

To answer these questions, we randomly selected 5,000 samples from the *Base Pool*, a collection of domain-specific queries, and fed them into the various models. We extracted the hidden state activations from each layer, focusing on the last token in the input query. This process was repeated for all layers of the models, ensuring a comprehensive analysis of the hidden state behavior. As a safety check, we introduced 5,000 samples from the *Specialized Pool*, which contained queries from a different distribution with no overlap with the MMLU dataset. This helped us ensure that the observed patterns were not merely the result of similar instructions or query semantics. Table 1 provides examples of the prompt variations across these datasets.

Figure 2 summarizes these traces' behavior for various LLM architectures, sorted by model size. The traces are color-coded by domains (Maths, Biomedical, Law, Humanities). We replaced the variance with standard deviation computation to use the same units as the original data, making it easier to relate the measure of dispersion back to the activation scale.[4] Semi-transparent lines (━)

---

[3]The queries were randomly selected, but we filtered out those with fewer than 10 tokens to remove queries that did not provide sufficient context (e.g. "Copyright © 2016 by").

[4]We omitted the mean activations because they were less stable than the standard deviation across different LLMs during experimentation.

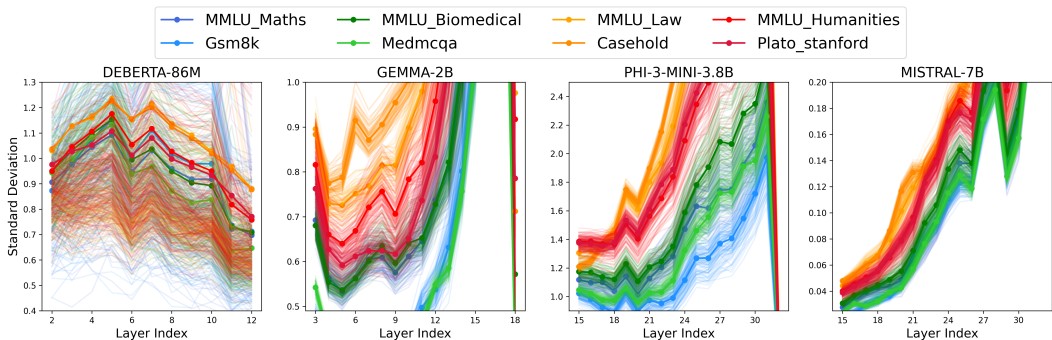

Figure 2: Standard deviation traces per datasets and samples across four different domains. Each subplot represents the behavior across layers $l$ on a different LLM architecture for MMLU, GSM8K, MEDMCQA, CaseHOLD, and PLATO datasets. Across all subplots, there is a general trend of increasing standard deviation in deeper layers, suggesting that as models progress through layers, the hidden states become more sensitive to the specific characteristics of each dataset. Further results for Llama-2B model are reported in Appendix A.3.

indicate the standard deviation of the raw hidden states for each independent random sample drawn from the *Base and Specialized Pools*, while bold lines with markers (⬤) show the aggregate standard deviation across all samples in each domain.

From our analysis, we identified several key trends:

- **Absence of pattern in DeBERTa:** The traces produced by the DeBERTa model did not exhibit a clear pattern, unlike autoregressive LLMs. This may be attributed to its bidirectional encoding architecture, which integrates left and right context, leading to less predictable activation patterns compared to autoregressive models that rely on sequential context. Also, the hidden states are more representative of the semantic and positional space in encoder architectures, which are optimized for tasks like classification, question answering, and sentence representation.

- **Consistency across generative LLMs:** The hidden state traces in autoregressive models showed consistent clustering around domain-specific queries. When samples from the *Specialized Pool* were introduced, a clear separation between domain-related queries emerged, indicating that these models are capable of distinguishing between domain-related requests beyond simple semantic similarities.

- **Data-dependent variability:** Across all models, the hidden state traces showed a consistent variance pattern, suggesting that the differences in behavior were not dependent on the model architecture, but rather were tied to the inherent characteristics of the datasets.

Appendix A.4 further explores the behavior of fine-tuned versions of Phi-3-mini-3.8B and Llama2-7B models. The hidden states separation remained largely unchanged post fine-tuning, suggesting that the domain-specific traces are properties of the pretrained models that persist even after task-specific fine-tuning, as the fine-tuned models are trained for a much shorter time than the pretrained models.

## 5.2 CONSISTENCY ACROSS PROMPT STYLES

To test the consistency of the traces against perturbations of the prompt, we constructed a new setup consisting of three new *Domain-Related pools*. These pools consist of samples from a variety of datasets within three domains: Medical[5], Maths[6], and Law[7]. We applied multiple prompt templates

---

[5]The Medical pool contains 16,711 samples from MMLU Biomedical, MEDMCQA (Pal et al., 2022), USMLE (Jin et al., 2020), and PubmedQA (Jin et al., 2019) datasets.

[6]The Maths pool contains 12,383 samples from MMLU Maths, GSM8k (Cobbe et al., 2021), OrcaMath (Mitra et al., 2024), and Math (Hendrycks et al., 2021b) datasets.

[7]The Law pool contains 11,712 samples from MMLU Law, CaseHOLD (Zheng et al., 2021), Scotus and Eurlex from LexGLUE benchmark (Chalkidis et al., 2022).

Table 1: Prompt Templates utilized for the *Maths Pool*. The instruction templates differ from closed to open instructions. In some (uniformly random) cases the chat template is omitted to make the task more challenging, enabling observation of how the trace deviates when no context guidance is provided.

| Source | Prompt Templates Example |
|---|---|
| MMLU Maths | Answer the following question: Up to isomorphism, how many additive abelian groups G of order 16 have the property that x + x + x + x = 0 for each x in G ? Options: A) 0 B) 1 C) 2 D) 3 Answer: |
| GSM8K | Q: A robe takes 2 bolts of blue fiber and half that much white fiber. How many bolts in total does it take? A: |
| Orca Math | A number divided by 10 is 6. Yoongi got the result by subtracting 15 from a certain number. What is the result he got? |
| Math | Answer the following question in the format \\boxed{answer} QUESTION: $$\frac{\sin^4 x + \cos^4 x - 1}{\sin^6 x + \cos^6 x - 1}.$$ FULL ANSWER: |

(detailed in Table 1 and Appendix A.7 for the Maths Pool and the two other domains respectively) to assess whether the traces deviate with changes in prompt structure.

Figure 3 demonstrates the variation in hidden state traces for the Phi-3-mini-3.8B model across different prompts and datasets. Our analysis reveals the following:

- **Prompt sensitivity in early layers:** The hidden state traces showed some variation in the early layers (up to layer 16), particularly in the Law domain. This suggests that some domains are more context-sensitive in order to generate their responses.
- **Stable representations in deeper layers:** From layer 16 onward, the traces stabilize across different prompts, indicating that the deeper layers are responsible for maintaining domain-specific representations, even in the presence of prompt perturbations.

This aligns with previous research (Meng et al., 2022) that suggests that early layers handle the input's structural and semantic properties, while the middle layers map facts and the last layer generalizes the output.[8] These findings suggest that LLM hidden state traces offer a robust representation of domain-specific information, which is largely invariant to prompt style changes. This stability makes hidden state-based representations a promising tool for understanding domain context, that can be extended to a variety of applications such as cross-domain model selection.

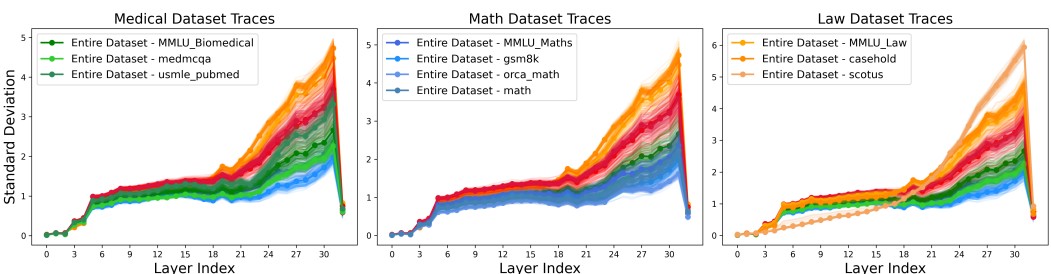

Figure 3: Standard deviation of the hidden state traces of Phi-3-mini-3.8B across 12 data sources and different prompt instructions for the domains of Maths, Biomedical, and Law. Each subplot contains the traces from 3-4 different datasets distributions belonging to the same domain. The legends in each subplot correspond to each dataset used for evaluation. Appendix A.6 provides the traces across the same datasets for Gemma-2B and Mistral-7B model, showing that this behavior is reproducible across other LLM families.

## 5.3 BENCHMARK WITH TRADITIONAL METHODS

To leverage these *domain-related* traces, we created the following setup:

---

[8]It is important to note that the traces could be the result of the injection of multiple relationships represented by the queries. However, we have observed that one of these relationships can be established as domain-related trajectories.

1. We selected three fine-tuned versions of Phi-3-mini-3.8B on the following subdomains: Emotional, Mathematical thinking and Medical Data. Detailed information about these checkpoints can be found in Appendix A.4. For simplicity, we refer to these models as *Phi-3-Domain*.

2. We conducted zero-shot evaluations on the finetuned checkpoints using the `lm_eval` library (Gao et al., 2024) on samples from the *Base Pool* – since these traces are known to exhibit similar behavior. This allows us to verify that each finetuned model is indeed proficient for its respective domain.

3. We map the "best-model" (i.e., the model with the highest performance on the benchmark) for each trajectory as follows:
   - Maths → Phi-3-MATHS
   - Biomedical → Phi-3-MEDICAL
   - Law and Humanities → Phi-3-PRETRAINED

4. We use a Multi-Layer Perceptron (MLP) to process the generated hidden states and learn to discriminate between domain traces. *Semantic Layer* and *DeBERTa classifier* are compared on the same task. Details of the MLP implementation can be found in Appendix A.2.

We trained the MLP classifier using raw hidden state traces from 4,000 random samples of the *Base Pool*. The training used a learning rate of 1e-4 with the Adam optimizer and a weight decay of 1e-2 over 3 epochs. Note that the same training data was used for all methods to maintain consistency in evaluation. Additionally, we included the *LLM Sequence Classifier* baseline, which relies on the complete prefill + generation process, to compare with the *LLM Hidden States Classifier*. This method, though more expensive due to multiple forward passes, offered a useful reference for evaluating the full input analysis capability.

We selected a subset of 5 different datasets (not seeing during training) to compare the final zero-shot performance of each method, results are reported in Table 2. The *LLM Hidden States Classifier* consistently outperforms the fine-tuned models and other baselines, particularly in open-ended tasks like GSM8K and domain-specific tasks like MEDMCQA.

Table 2: Routing performance is measured by task accuracy, with each sample dynamically assigned to a preferred model for evaluation by the routing mechanism. The *Domain fine-tuned* baseline refers to the model that showed the best performance in the initial domain test dataset. The *LLM Hidden States Classifier* achieves the highest overall improvement, outperforming domain fine-tuned models in several cases.

| | MMLU | GSM8K | MATH | MEDMCQA | USMLE | CaseHOLD | Avg Acc | % Imp |
|---|---|---|---|---|---|---|---|---|
| Domain fine-tuned | **0.683** | 0.400 | 0.057 | 0.258 | 0.228 | 0.487 | 0.352 | |
| LLM Hidden States Classifier | 0.665 | **0.560** | **0.144** | **0.270** | 0.241 | **0.492** | **0.395** | **+12.3%** |
| DeBERTa Sequence Classifier | 0.668 | 0.395 | 0.060 | 0.261 | 0.228 | 0.487 | 0.350 | **-0.7%** |
| Semantic Router | 0.658 | 0.374 | 0.064 | 0.248 | **0.255** | 0.480 | 0.336 | **-9.2%** |
| DeBERTa Hidden States Classifier | 0.630 | 0.183 | 0.086 | 0.243 | **0.255** | 0.480 | 0.313 | **-11.2%** |
| LLM Sequence Classifier | 0.648 | 0.118 | 0.071 | 0.257 | 0.232 | 0.480 | 0.302 | **-14.4%** |

We use the *mapped* model as the main baseline in Table 2, i.e., the model that performs best within each domain. However, the results show that the *LLM Hidden States Classifier* consistently improves overall performance, outperforming both the semantic layer and DeBERTa encoder methods, which do not match the baseline performance.

Interestingly, the *Hidden States Classifier* performs better than the domain fine-tuned models in several cases. This may seem counterintuitive, as one might expect a fine-tuned model to excel in the specific domain it was trained on. This discrepancy could be due to the fine-tuned models overfitting to the characteristics of their training datasets, thereby missing the generalization capabilities needed for cross-domain tasks. Also, the hidden states capture richer representations of domain-specific trajectories, allowing for better cross-domain generalization.

In some cases, however, the decrease in performance is worse. As a safety check, we trained the same MLP classifier on the hidden states extracted from the DeBERTa encoder model instead of Phi-3-mini-3.8B. The DeBERTa model performs poorly; this is expected since, as shown in Figure 2,

the hidden states from DeBERTa do not exhibit the same clear patterns across domains, which likely explains its lower effectiveness in this task. This suggests that autoregressive models, like Phi-3-mini-3.8B, are better suited for capturing domain-related trajectories in hidden states, while encoder models such as DeBERTa, which focus on bidirectional context, might not generate domain-specific traces in the same way.

We also compared the *LLM Sequence Classifier* and the *LLM Hidden States Classifier*. The results indicate that separating the analysis of the input from the generation phase (as done in the hidden states approach) leads to more robust representations. By leveraging the entire sequence of hidden states, the *LLM Hidden States Classifier* captures more detailed information about the input sequence, improving its ability to make accurate predictions. Additionally, in the generation process, hidden states are influenced by previous tokens, which can narrow the representation of domain information. In contrast, the prefill phase retains the rich, diverse embeddings from the pretrained model, offering a more flexible and unbiased understanding of the domain.

In summary, our results show that moving away from rigid domain-labeled model selection strategies toward approaches that rely on hidden state representations can lead to improved generalization across domains, questions, and input structures.

### 5.4 TRADEOFFS OF REDUCING LAYERS COMPUTATION.

Reducing the number of hidden layers fed to the MLP classifier can help reducing latency and computational costs during inference. Therefore, we investigate how performance evolves as we progressively reduce the number of layers. [9]

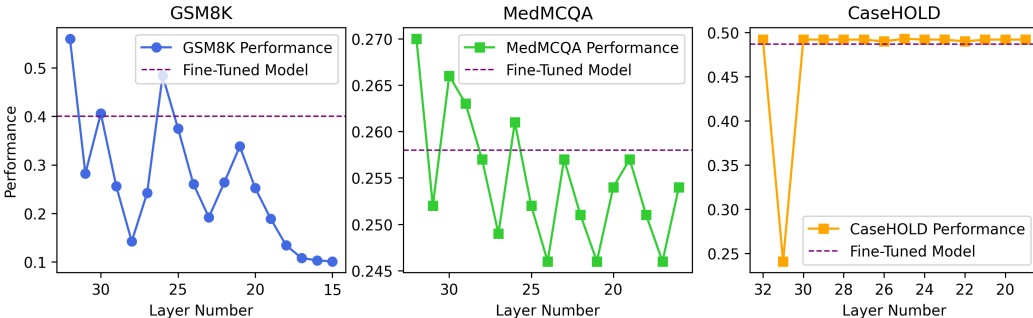

Figure 4: Zero-shot Accuracy Performance as we are reducing the number of layers used in the MLP discriminator, for open-ended (GSM8k) and multichoice (MEDMCQA, CaseHOLD) tasks. Each point in the subplots is cumulative, incorporating signals from layers 1 to $X$.

For these experiments, we replicated the setup described in the previous subsection while varying the number of layers fed to in the MLP classifier for training and inference. The results in Figure 4 show that layer 26 is the turning point where the hidden states unlock the ability to improve the performance of the fine-tuned model. However, the best performance is obtained by computing all 32 layers. This can be justified with previous observation that later layers in the model maintain domain-specific representations, therefore, some "incomplete" representations can cause the drop in performance for some tasks on layers 26-32.

The drop in performance is greater for the GSM8k task, which requires open-ended generation and verifying the exact-match answer after the model develops the Chain-of-Thought. Similarly, in Table 2, we observe that the highest performance improvement comes from the GSM8k and MATH datasets, both of which belong to the same open-ended generation category.

## 6 LIMITATIONS

While our study provides valuable insights into the utility of hidden state representations in LLMs, several limitations should be acknowledged:

---

[9]The prefill phase is not autoregressive. Then, finding an optimal layer means that we need to compute all the layers up to that point.

- **Focus on smaller LMs**: Our work primarily focuses on interpreting small LLMs (up to 7B parameters). While we demonstrate that even the smallest models can provide useful information on domain interpretation, the applicability of our approach to larger models remains to be explored.

- **Domain traces may not generalize**: The domain traces we show per model may not be a definitive representation of that domain, but rather an infusion of multiple subdomains reflected in the queries. Therefore, the "clustering" might not generalize to other datasets sharing the same domain label. We argue, however, that this variability reflects how the model internalizes new data distributions. This characteristic represents a key advantage of using hidden states for interpretation, as it allows us to gain feedback directly from the model itself.

## 7 DISCUSSION

When creating strategies for interpreting domain questions, it is more beneficial to focus on the model's comprehension of the question itself rather than the domain labels. This approach can lead to a variety of advantages and interesting scenarios. In this paper, we aimed to uncover the LLM's ability to differentiate between well-defined domains and leverage the context understanding into domain representations that can be harnessed in the model routing scenario, showing an improvement of 12% over baseline methods. However the applicability of this approach can be extended to tasks with:

- **Interdisciplinary collaboration:** For instance, biomedical ethics, where questions involve ethical reasoning but also medical knowledge, selecting models or agents, based on their interpretation of the question's complexity or reasoning requirements rather than a domain label can improve performance.

- **Unsupervised model selection:** In the absence of labeled data, selecting models based on their ability to interpret the structure or type of reasoning involved can be helpful in zero-shot learning tasks. Models can be chosen that are good at recognizing the tone or domain of the question, which can be more beneficial than relying on static routing to domain fine-tuned models.

- **Remove manual bottleneck:** These mechanisms can be leveraged to enhance scalability by eliminating the bottleneck associated with manual sample selection, thereby streamlining the processing of large datasets on well-defined domains.

- **Enhancing LLM-Human collaboration:** The domain representations can be harnessed to generate summaries or feedback of the LLM context understanding when there is uncertainty on how the model would process an specific request.

We have demonstrated that LLMs are capable of encoding domain representation, capturing contextual information in their hidden states –before the generation phase– distinguishing between queries from different domains, regardless of the prompt style and query source. This approach is particularly useful in domains where the labels are insufficient to capture the complexity of the underlying data. Our approach can be used to identify the most relevant model for a given *domain-related trajectory* and improve results over semantic and token-based approaches. Comparing these three methods has provided us with new insights into their strengths and limitations, which can be useful for future research in this area. Our approach shows promise for improving the interpretability of language models, which will hopefully lead to a better understanding of their underlying mechanisms and discriminative power.

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

# A APPENDIX

## A.1 SUBCATEGORIES USED FOR THE MMLU DATASET

For the initial evaluation on the MMLU Dataset we subsampled 30 random categories from the complete test set. We followed the domain labeling provided by the dataset authors in the Github Repository https://github.com/hendrycks/test/blob/master/categories.py, to provide a better categorization of the different samples as shown in Table 3.

Table 3: MMLU Dataset original subcategories turned into 4 domains for the *Base Pool*.

| Domain Category | Original MMLU Subcategory | Samples |
|---|---|---|
| **Maths and Logical** | abstract_algebra
college_mathematics
elementary_mathematics
high_school_mathematics
high_school_statistics | 1064 |
| **Biology / Chemistry / Health** | anatomy
college_biology
high_school_biology
human_aging
human_sexuality
medical_genetics
nutrition
virology
clinical_knowledge
college_medicine
professional_medicine
college_chemistry
high_school_chemistry | 2528 |
| **Law** | international_law
professional_law
jurisprudence | 1763 |
| **Humanities** | high_school_european_history
high_school_us_history
high_school_world_history
prehistory
formal_logic
logical_fallacies
philosophy
world_religions | 2003 |

## A.2 MLP CLASSIFIER USED FOR HIDDEN STATES TRAJECTORIES

We employed a Multi-Layer Perceptron (MLP) as a classifier to process the hidden states generated by Phi-3-mini-128k. The MLP is structured with three fully connected layers. The input layer, which takes the hidden states, is followed by two hidden layers. The first fully connected layer (fc1) maps the input to a hidden dimension of size hidden_size using a linear transformation, followed by a ReLU activation function. The output of the first hidden layer is then passed through a second fully connected layer (fc2), which retains the same hidden dimension, again followed by a ReLU activation. The final layer (fc3) maps the hidden representation to the output space, producing a prediction over 4 classes.

## A.3 LLAMA-2B HIDDEN STATES ANALYSIS

Figure 5 presents the standard deviation calculated from the raw hidden states of the Llama2-7B model. Unlike the architectures shown in Figure 2, the *domain-trajectories* here appear to fall

within similar ranges at first glance (left subplot), with the exception of the law domain datasets, which exhibit more variability in standard deviation across most layers. However, a closer look (right subplot) reveals distinct differences in the colors representing each domain. This observation suggests that the Llama2-7B model may encode domain-specific information in a more nuanced manner.

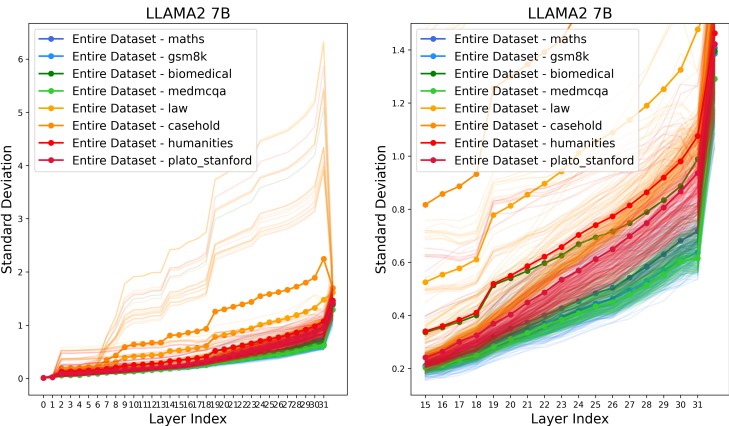

Figure 5: Standard deviation traces per datasets and samples across four different domains, extracted from Llama2-7B model. The law domain datasets, in particular, stand out with their higher variability, indicating that the model's hidden states are more sensitive to the specific characteristics of legal texts - which is a similar behavir presented as Phi-3-mini-3.8B in Figure 3. This nuanced encoding could be a result of the model's training data.

### A.4 TRACES MIGHT REMAIN AFTER FINE-TUNING

We used some of the public checkpoints that were already pretrained for the Phi-3-mini-3.8B and Llama2-7B models that are available at Huggingface. Our aim was to test how much the original traces across activations in the pretrained model changes once it has been fine-tuned for different domains. The description of each checkpoint that we utilized is given below.

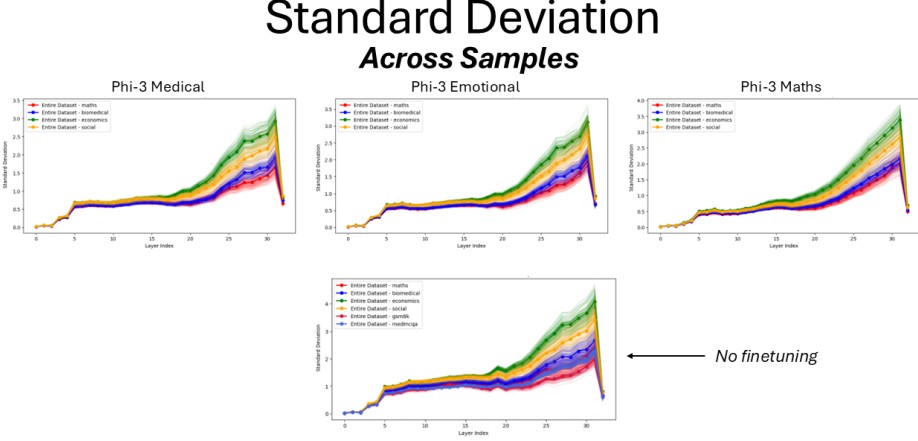

Figure 6: Standard Deviation of Phi-3-mini-3.8B across different fine-tuned versions.However, it is worth noting that the emotional and medical versions appear to be a scaling of the original pretrained model. It should be noted that the finetuning process was not controlled, so no catastrophic forgetting was performed on purpose. Despite this, the results suggest that the model is robust and can be fine-tuned without significant changes to the original architecture.

1. Phi-3 Pretrained: `microsoft/Phi-3-mini-128k-instruct`

2. Phi-3 Maths: `dbands/Phi-3-mini-4k-instruct-orca-math-word-problems-200k-model-16bit`

3. Phi-3 Medical: `ChenWeiLi/MedPhi-3-mini_v1`

4. Phi-3 Emotional: `Evortex/EMO-phi-128k`

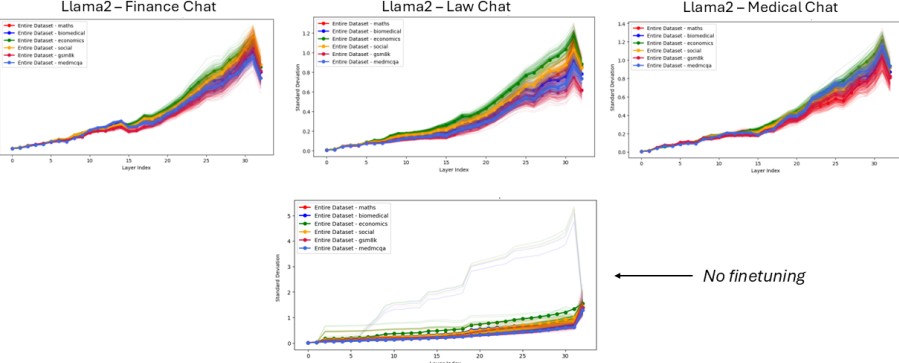

Figure 7: Standard Deviation of Llama Chat model. In contrast with the behavior observed in smaller models, we can see that Llama model keeps capturing the nuances for the Finance and Law versions. However, the Medical version has more overlapping across domains.

1. Llama2 Pretrained: `meta-llama/Llama-2-7b-chat-hf`

2. Llama2 Finance Chat: `AdaptLLM/finance-chat`

3. Llama2 Law Chat: `AdaptLLM/law-chat`

4. Llama2 Medical Chat: `AdaptLLM/medicine-chat`

## A.5 OVERLAPPING ACROSS MATHS AND BIOMEDICAL DOMAINS

The overlap in hidden states when computing queries from the mathematical and biomedical domains contrasts with domains like law and humanities, where reasoning processes differ. Math and biomedicine rely heavily on structured, logical reasoning and problem-solving, leading to more precise, analytical neural activations. In contrast, law and humanities emphasize interpretative, narrative-driven reasoning, which involves greater flexibility, ambiguity, and context-dependent thinking. While math and biomedical domains focus on clear relationships between variables and technical language, law and humanities require models to capture complex human experiences, ethical considerations, and persuasive argumentation. As a result, the hidden states for law and humanities queries would likely reflect more diverse and abstract linguistic patterns, with less direct overlap compared to the more systematic reasoning used in mathematics and biomedicine.

## A.6 PROMPT VARIATION ACROSS LLMS

Below we present further results on how Gemma-2B and Mistral-7B reflect the prompts variation across the different datasets and instructions. The prompt instructions utilized per each dataset are presented in Appendix A.7. For both architectures we can observe that the different instructions do not affect the general shape of the traces on each domain.

## A.7 PROMPT TEMPLATES UTILIZED FOR EACH DOMAIN-RELATED POOL

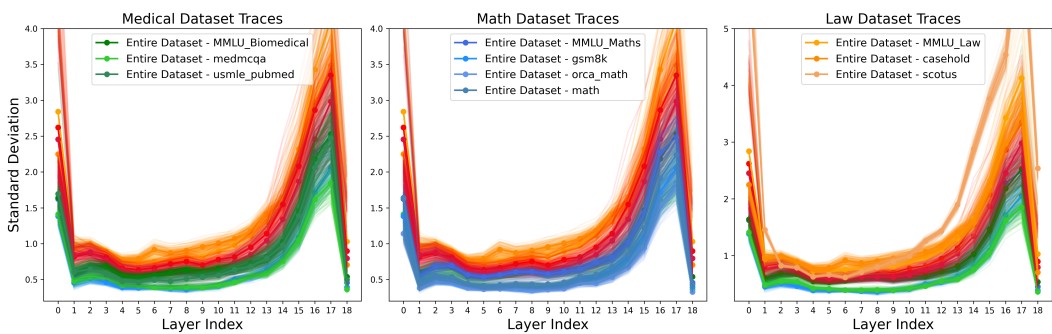

Figure 8: Standard Deviation computed on raw hidden states from Gemma-2B model. We inputted samples from 12 different datasets to the model, ensuring different prompts and distribution. Gemma-2B presents a bigger overlapping between medical and mathematical domains, meaning that the model characterizes these datasets very similarly and therefore from the raw hidden states it is more difficult to distinguish between these differences.

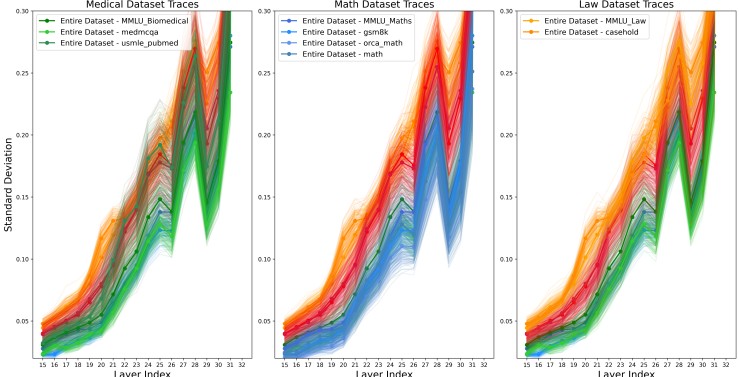

Figure 9: Standard Deviation computed on raw hidden states from Mistral-7B model. We inputted samples from 12 different datasets to the model, ensuring different prompts and distribution. We can observe that the domains characterization is preserved across the second half of the layers, noticing an overlapping between maths and medical domain as in previous architectures.

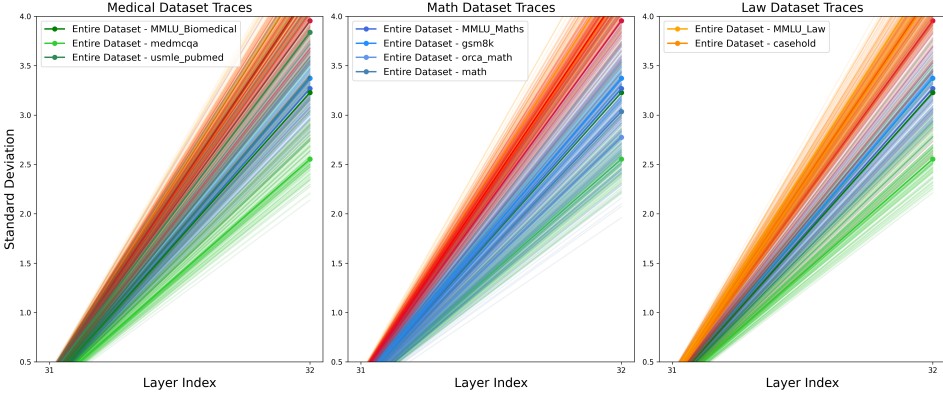

Figure 10: Zoom-in on the last layer of Mistral-7B traces in Figure 9.

Table 4: Prompt Templates utilized for the *Medical Pool*. The instruction templates differ from closed to open instructions in order to inspect whether the activation trace deviates from the original "sketch".

| Source | Prompt Templates Example |
|---|---|
| MMLU Biomedical | Answer the following question: A 37-year-old woman with right lower extremity edema is evaluated because of the sudden onset of shortness of breath and pleuritic chest pain. A diagnosis of pulmonary embolism is made. Which of the following signs, if present on physical examination, would be the most specific indicator of pulmonary arterial hypertension in this patient? Options: A) Increased jugular venous pressure B) P2 louder than A2 C) Peripheral edema D) Presence of an S3 Answer: |
| MEDMCQA | Select the best option for the following question: Axonal transport is: Options: 0) Antegrade 1) Retrograde 2) Antegrade and retrograde 3) None |
| USMLE | A 39-year-old man presents to the emergency department because of progressively worsening chest pain and nausea that started at a local bar 30 minutes prior. The pain radiates to the epigastric area. He has a 5-year history of untreated hypertension. He has smoked 1 pack of cigarettes daily for the past 5 years and started abusing cocaine 2 weeks before his emergency room visit. The patient is diaphoretic and in marked distress. What should be the first step in management? |
| PubMED | Are group 2 innate lymphoid cells ( ILC2s ) increased in chronic rhinosinusitis with nasal polyps or eosinophilia? |

Table 5: Prompt Templates utilized for the *Law Pool*. Similarly to the other domain-related pools, the instruction templates differ from closed to open instructions.

| Source | Prompt Templates Example |
|---|---|
| MMLU Law | Answer the following question: A resident announced his candidacy for state representative. A law in the state requires new political entrants (regardless of party affiliation) to obtain three times the number of signatures as other candidates who have run for office previously. The resident, however, failed to obtain the necessary number of authenticating signatures to have his name placed on the ballot. The resident filed a complaint in federal district court alleging the unconstitutionality of the authenticating requirement. Which of the following, if established, is the state's strongest argument for sustaining the validity of the authenticating requirement? Options: A) The resident's petition contained a large number of false signatures. B) A similar authenticating statute was held to be constitutional in another state the previous year. C) The authenticating requirement was necessary to further a compelling state interest. D) Two other candidates had successfully petitioned to have their names included on the ballot. Answer: |
| CaseHOLD | Your task is to complete the following excerpt from a US court opinion: § 3583(e)(3) was reasonably foreseeable and provided the defendant with a fair warning. Thus, it was not unconstitutional to apply Johnson retroactively. Although Seals is unpublished, and thus not binding, Seals is authoritative and persuasive. Therefore, applying Johnson retroactively to Martinez's 1993 conviction does not violate the Due Process Clause, and the district court did not plainly err in reimposing supervised release after the first revocation. Accordingly, Martinez's sentence is affirmed. AFFIRMED; MOTION DISMISSED AS MOOT. 1 . See, e.g., United States v. Golding, 739 F.2d 183, 184 (5th Cir.1984). 2 . Ketchum v. Gulf Oil Corp., 798 F.2d 159, 162 (5th Cir.1986). 3 . See Eberhart v. United States, 546 U.S. 12, 126 S.Ct. 403, 406-07, 163 L.Ed.2d 14 (2005) (per curiam) (holding that the defendants evidence did not qualify as newly discovered evidence |
| Scotus | 509 U.S. 418 113 S.Ct. 2696 125 L.Ed.2d 345 UNITED STATES and Federal Communications Commission, Petitioners,v.EDGE BROADCASTING COMPANY T/A Power 94. No. 92-486. Argued April 21, 1993. Decided June 25, 1993. Syllabus * Congress has enacted federal lottery legislation to assist States in their efforts to control this form of gambling. Among other things, the scheme generally prohibits the broadcast of any lottery advertisements, 18 U.S.C. § 1304, but allows broadcasters to advertise state-run lotteries on stations licensed to a State which conducts such lotteries, § 1307. This exemption was enacted to accommodate the operation of legally authorized state-run lotteries consistent with continued federal protection to nonlottery States' policies. North Carolina is a nonlottery State, while Virginia sponsors a lottery. Respondent broadcaster (Edge) owns and operates a radio station licensed by the Federal Communications Commission to serve a North Carolina community, and it broadcasts from near the Virginia–North Carolina border. Over 90% of its listeners are in Virginia, but the remaining listeners live in nine North Carolina counties. Wishing to broadcast Virginia lottery advertisements, Edge filed this action, alleging that, as applied to it, the restriction violated the First Amendment and the Equal Protection Clause. The District Court assessed the restriction under the four-factor test for commercial speech set forth in Central Hudson Gas & Electric Corp. v. Public Service Comm'n of New York, 447 U.S. 557, 566, 100 S.Ct. 2343, 2351, 65 L.Ed.2d 341|(1) whether the speech concerns lawful activity and is not misleading and (2) whether the asserted governmental interest is substantial; and if so, (3) whether the regulation directly advances the asserted interest and (4) whether it is not more extensive than is necessary to serve the interest concluding that the statutes, as applied to Edge, did not directly advance the asserted governmental interest. The Court of Appeals affirmed. Held: The judgment is reversed. 956 F.2d 263 (CA 4 1992), reversed. Justice WHITE delivered the opinion of the Court as to all but Part III-D, concluding that the statutes regulate commercial speech in a manner that does not violate the First Amendment. Pp. ___ |

