# OpenReview forum: "Exploring How LLMs Capture and Represent Domain-Specific Knowledge"
_ICLR.cc/2025/Conference — Submitted to ICLR 2025_

### Official Review · Reviewer_xcr7 · 2024-11-01

**Soundness:** 3
**Presentation:** 2
**Contribution:** 3
**Rating:** 5
**Confidence:** 3

**Summary:**

This study examines whether Large Language Models (LLMs) can recognize domain-specific nuances in natural language queries by analyzing its states, revealing that LLMs can distinguish queries from different domains. The findings suggest that LLMs can differentiate related domains and that the best-performing model isn't always the fine-tuned one, with applications to both closed and open-ended generative tasks. This study includes 4 LLM models ranging 2B-7B parameters and a subset of MMLU dataset

**Strengths:**

The paper addresses a very interesting topic in unfolding how LLMs works and can be utilized and selected for various tasks.
The chosen dataset is appropriate and also the four chosen models.
The work is very interesting, in this new era of LLMs and a focus on transparency and trust of such models. The quality of the work is good, given the approach, experiments, and baselines considered, but can be improved.
The paper presentation can also be improved, as it is highlighted below.

**Weaknesses:**

- i would have liked a more comprehensive discussion on the dataset, the decision to use 30 / 57 domains and also a characterization of the various subdomains, instead of mentioning just the 4 ones the paper decided to focus on.
- Next steps / research directions are unclear
- it would have been appropriate a discussion on computational runtime in these studies
- Some appendix material should be added to the core paper: in particular the discussion in A.3, A.4, A.5

Some other notes:
- the a/b/c/d labels should be clear by domain/sample
- a diagram showing the models (in/out and internals) and approach would have been good for a clearer presentation
- Figure 4: "Performance" label is unclear

**Questions:**

- Could you explain the rationale behind selecting 30 out of 57 domains? Additionally, can you provide a brief characterization of the various dataset subdomains, perhaps in an appendix?
- can you please clarify what other practical applications do you envision for this method? for what purpose / practical application? Routing Strategies is one mentioned, but a more detailed discussion is warranted on concrete examples of how it might be implemented in real-world scenarios.
- what are the runtime execution of such experiments? and what would be for a much bigger model?
- line 223: the baseline method are carefully explained, but not the rationale behind these choice. can you please clarify how you chose such parameters?
- Could you provide details on the computational resources used and the runtime for your experiments? How might these scale with larger models or datasets?
- What future research directions do you envision based on these findings? Are there particular applications or extensions of this work that you think would be most promising to explore next?

---

> ### Author Response · Authors · 2024-11-22
> **Authors Response (Part 1)**
>
> We sincerely thank the reviewer for their insightful comments and constructive feedback. Below, we address each point raised.
>
> **Dataset and domain characterization:** The decision to select 30 out of the 57 domains from the MMLU dataset was driven by the need to focus on specific domains where the required skills for LLMs differ significantly, providing a clearer spectrum for analyzing contrastive behavior. This approach was inspired by the AdaptLLM [1] paper, which demonstrated diverse skill requirements across domains such as Biomedicine, Finance, and Law. Our comparative analysis of models based on the Llama2 backbone, included in Figure 7, builds on these models.
>
> Additionally, we included the mathematical domain due to their large sample size within the MMLU dataset and the abundance of publicly available fine-tuned checkpoints. This allows for broader comparisons and a more comprehensive evaluation. Ultimately, 30 subdomains qualified under these domain categories, as detailed in the GitHub repository of the dataset and listed in Appendix A.1. The remaining subcategories, such as those categorized under miscellaneous or global facts, were excluded to avoid ambiguity and ensure clearer, more interpretable results.
>
> It is important to note that only the MMLU dataset underwent this filtering process; the other eight datasets were utilized in their entirety without filtering any samples. We thank the reviewer for their valuable feedback and will enhance Appendix A.1 to include a detailed list of the selected subdomains, along with additional information about their content.
>
> **Next steps/ Research directions:** We envision that the findings from analyzing domain-specific hidden state patterns can have several practical applications:
>
> _Steering Model Behavior:_ By modifying hidden states in real time along “well-known” trajectories, it becomes possible to guide the model’s outputs toward specific behaviors or styles. For example, steering vectors can be employed to influence the generation process within a specific domain, enhancing the model’s adaptability.
>
> _Data/Training Efficiency:_ Hidden state traces can serve as a proxy for identifying features that drive faster model convergence, facilitating more efficient use of training data. This concept aligns with ideas like the “Lottery Tickets Dataset,” enabling optimization of data utilization and reducing training costs.
>
> _Sparse Model Optimization:_ In our recent experiments with the Phi-3.5-MoE model, we observed that sparse models exhibit predictable activation patterns in the early layers of their hidden states. These patterns could potentially allow for early predictions of which experts will be activated, paving the way for dynamic load balancing and smarter compute allocation.
>
> These examples represent just a few promising avenues for further exploration. While they extend beyond the scope of this paper, they highlight significant opportunities for future research. We will update the final discussion to better articulate these potential directions and provide additional clarity.
>
> **Regarding appendix A.3, A.4 and A.5:** Due to space constraints, we decided to relocate this content to the appendix to ensure sufficient focus on highlighting the main contributions of this work. To maintain accessibility, we have added footnotes in the main text with clear pointers to the corresponding content in the appendix. We will ensure this is further emphasized in the final version for improved clarity and navigation.
>
> **Regarding hyperparameter selection for baseline method:** For hyperparameter selection across all baseline methods, we ensured the best achievable performance for each respective model while using the same training samples across all experiments. This approach allowed us to maintain a fair comparison under consistent conditions, ensuring an “apples-to-apples” evaluation framework.

---

> > ### Author Response · Authors · 2024-11-22
> > **Authors Response (Part 2)**
> >
> > **Computational Runtime and Resource Scaling:** As outlined in Section 4, the experiments were conducted using three NVIDIA RTX A6000 GPUs, each with 44 GB of memory. The memory requirements for inference scaled approximately with the number of parameters in each model. While larger models demand more computational resources, we emphasize that future applications may not require larger models if smaller models can produce similar behavioral insights.
> > The runtime also scales with the number of datasets used, but the latency remains minimal compared to performing multiple forward passes during the generation phase. The computational cost primarily lies in the prefill phase, which is both less resource-intensive and capable of effective parallelization.
> >
> > To provide further clarity, the following table outlines the average latency per sample for various methods tested on 1,000 MMLU samples (Table 2). While the LLM Hidden States Classifier (running only the prefill phase of Phi-3-mini) demonstrates higher latency (in seconds) than the DeBERTa Sequence Classifier, this difference can be reduced by decreasing the number of layers required per domain.
> >
> > | Method                          | Eval Avg Latency (MMLU-1k samples) |
> > |---------------------------------|------------------------------------|
> > | LLM Hidden States Classifier    | 0.366                              |
> > | DeBERTa Sequence Classifier     | 0.160                              |
> > | Semantic Router                 | 0.040                              |
> > | DeBERTa Hidden States Classifier| 0.037                              |
> >
> > We appreciate the reviewer’s attention to these aspects and are happy to incorporate additional latency details in the final version of the paper to provide a comprehensive understanding of resource efficiency.
> >
> > [1] Adapting Large Language Models to Domains via Reading Comprehension

---

### Official Review · Reviewer_2yn5 · 2024-11-02

**Soundness:** 4
**Presentation:** 3
**Contribution:** 3
**Rating:** 5
**Confidence:** 4

**Summary:**

This paper presents quite an intriguing investigation into how LLMs encode domain-specific knowledge. The authors have undertaken a comprehensive study examining hidden state patterns during the prefill phase, introducing what they've termed "latent domain-related trajectories." Their experimental work spans multiple architectures - Gemma-2B, Phi-3-mini-3.8B, Llama2-7B, and Mistral-7B - and demonstrates rather fascinating patterns in how these models process domain-specific queries. They've shown that hidden states can serve as reliable indicators of domain understanding, leading to a 12.3% improvement over baseline methods in model selection tasks. The work includes thorough analyses across various domains - medical, mathematical, and legal - and examines the robustness of these patterns across different prompt styles. Rather innovative, I must say, particularly in their approach to leveraging these patterns for practical applications in model selection and routing.

**Strengths:**

The authors have demonstrated remarkable creativity in their approach. The idea of examining hidden states for domain understanding is quite novel, and their experimental methodology shows careful consideration. I particularly appreciate their comprehensive evaluation across multiple architectures and domains. The practical implications for model selection could be quite significant, if properly developed.

**Weaknesses:**

1) In Section 2 and 3, the theoretical foundation appears inadequate. Authors discussed prior work but fail to establish a clear theoretical connection between hidden states and domain representation. The mathematical formulations lacks rigorous justification for why these specific activation patterns should correlate with domain knowledge. The "latent domain-related trajectories" introduced in L 68 need stronger mathematical grounding beyond empirical observations. Furthermore, the variance computation described in equations (1) and (2) requires proper statistical analysis of its significance in domain representation.

2) The experimental setup described in Section 4 reveals several critical issues. The model selection appears arbitrary, particularly regarding the choice of only testing models up to 7B parameters. The domain categorization described in lacks systematic justification for the grouping criteria. In Section 5.2, the prompt consistency analysis (L 327-337) needs more rigorous testing across a broader range of prompt variations. The performance improvements reported in Table 2 lack error bars and statistical significance testing, making it difficult to assess the reliability of the 12.3% improvement claim.

3) The implementation details in Section 5.3 require further clarification. The MLP classifier architecture described around lines 385-390 lacks proper justification for its design choices. The hyperparameter selection process mentioned in L 395-401 needs more detailed documentation. Most critically, the computational complexity analysis is entirely missing from Section 5.4, where the authors discuss layer reduction without properly quantifying the performance-computation tradeoffs.

**Questions:**

1) Can you provide mathematical proofs for why these trajectories should exist?
2) For Table 2: What is the statistical significance of the reported improvements?
3) How sensitive are your results to different prompt formulations?
4) Section 5.4: Can you quantify the computational savings from layer reduction?

---

> ### Author Response · Authors · 2024-11-22
> **Authors Response (Part 1)**
>
> We sincerely thank the reviewer for their insightful comments and constructive feedback. Below, we address each point raised.
>
> **Insufficient theoretical foundation:** We acknowledge that our study primarily focuses on empirical observations rather than presenting a fully developed theoretical framework. The central aim of this work is to experimentally investigate hidden state patterns to identify trends that suggest domain-specific processing by utilizing information from all layers.
>
> Regarding the "latent domain-related trajectories" introduced in the paper, we agree that their correlation with domain representation would benefit from more robust mathematical grounding. Our intention was to showcase consistent patterns across diverse domains—such as medical, legal, and mathematical—when analyzed across various architectures and datasets. These patterns, while not constituting definitive theoretical proof, provide preliminary evidence of the influence of domain knowledge on model intermediate activations on the prefill phase.
>
> We also appreciate the reviewer's suggestion for more rigorous statistical validation of the variance computations in equations (1) and (2). While our current analysis provides a measure of variability in activations linked to domain-specific tasks, we acknowledge the need for further justification. To address this, we are incorporating entropy across activations as an additional validation metric in our experiments for the final version. Entropy has been widely recognized as a robust measure for quantifying uncertainty and variability in model behavior, offering a complementary perspective to variance.
>
> **Model selection in Section 4:** Our decision to focus on smaller language models (below 7B parameters) was primarily driven by resource constraints, as experimenting with larger models demands significantly more computational resources. We acknowledge this limitation and have explicitly noted in Section 6 that the applicability of our approach to larger models remains an avenue for future investigation.
>
> We recently extended our experiments to include the Phi-3.5-MoE instruction-tuned model (41B parameters). These experiments revealed that sparse models exhibit distinct separation of traces from the early layers, offering exciting possibilities on other research areas such as dynamic expert allocation. These findings, which expand the scope of our conclusions, will be included in the final version appendix to provide further context.
>
> We respectfully disagree with the suggestion that our model selection seems arbitrary. Our approach was deliberately designed to validate our findings across a diverse range of widely-used open-source model architectures, training methodologies, and parameter sizes. This variety ensures that our conclusions are both robust and derived from varied experimental setups, enhancing the generalizability of our observations. Should the reviewer have any specific concerns regarding our model selection or require further clarification on any aspect, we would greatly appreciate the opportunity to address them.
>
> **Domain categorization lacks systematic justification:** The domain categorization for each dataset was determined based on the nature of the questions, with all datasets belonging to a single domain except for MMLU, whose categories are detailed in Appendix A.1, as explained in Section 4. We are uncertain about the specific aspects of our approach to domain categorization that you find unclear or lacking. Could you kindly elaborate on this point so that we may address your concerns more effectively?
>
> **Prompt consistency analysis:** In our experiments, we tested four different prompt variations per domain (see Tables 1, 4, and 5), while maintaining the core intent of each question. In some cases, we deliberately omitted context to create more challenging scenarios for the model. The results for these cases are shown in Figure 3, where we observe minimal variance across the last 10 layers of the baseline model. Additionally, we include variations across the fine-tuned models in Figure 6. In all cases, we observe that the model tends to group queries based on domain similarity.
>
> As noted in Table 2, each dataset has its own prompt variation (for example, although GSM8K and MATH are from the same domain, each dataset uses a specific variation, as detailed in Table 1). We believe we have sufficiently explored prompt variation in our setup, demonstrating robustness across different scenarios. However, we welcome any further suggestions on how we might enhance this analysis or explore additional prompt variations to strengthen our findings.

---

> > ### Author Response · Authors · 2024-11-22
> > **Authors Response (Part 2)**
> >
> > **Lack of error bars in Table 2:** We agree that error bars are crucial for better illustrating the variability and statistical significance of our findings. We will make sure to include them in the final version of the paper, improving the clarity and robustness of the presented data.
> >
> > **MLP architecture and hyperparameter selection:** We have provided a detailed explanation of the MLP layer in Appendix A.2., due to space constraints in the main paper. While we acknowledge that an MLP may not be the ideal architecture for this task, we chose it as a proof of concept to demonstrate the potential of extracting meaningful patterns from hidden state activations across layers. The MLP itself is not the primary focus of this paper but serves as an example of how such patterns can be utilized. We hope that this explanation in the appendix will provide further clarity on our approach.
> >
> > **Computational analysis/savings from layer reduction:**  We would like to clarify that the primary goal of Figure 4 is to investigate whether it is computationally efficient to reduce the number of layers needed to determine the domain of a sample. As shown in the figure, the reduction in computation is minimal, still requiring approximately 26 layers. Nonetheless, we believe it is valuable to include this finding in the main text for clarity. We will ensure the final numbers of latency are also included in the revised version of the paper.

---

### Official Review · Reviewer_uiqH · 2024-11-04

**Soundness:** 3
**Presentation:** 2
**Contribution:** 2
**Rating:** 5
**Confidence:** 3

**Summary:**

The paper investigates how Large Language Models (LLMs) capture domain-specific nuances by analyzing hidden states during the prefill phase. The authors introduce the concept of "latent domain-related trajectories," which reveal domain sensitivity. They claim that these trajectories provide a robust signal for domain-specific tasks and model selection, leading to improved performance in cross-domain generalization tasks like legal, medical, and mathematical reasoning.

**Strengths:**

* Originality: Proposes a novel method to use hidden state trajectories for domain-specific model selection.
* Quality: The experiments are comprehensive, covering various architectures and tasks.
* Significance: Potential utility in improving model selection.

**Weaknesses:**

* Clarity: The writing is not engaging, making the paper hard to follow.
* Justification: The rationale for why these hidden state patterns indicate domain sensitivity is weak.
* Generalizability: Limited applicability beyond the datasets studied, as acknowledged in the limitations.
* Interpretability: It’s unclear how this work significantly enhances our understanding of LLM behaviour or interpretability.r

**Questions:**

* Can you elaborate on why the observed hidden state patterns conclusively indicate domain-specific understanding?
* How do you envision this approach being practically used, given the need to process queries through multiple models before selection?

---

> ### Author Response · Authors · 2024-11-22
> **Authors Response**
>
> We sincerely thank the reviewer for their insightful comments and constructive feedback. Below, we address each point raised.
>
> **Regarding clarity of the paper:** We appreciate the reviewer’s feedback regarding the clarity and engagement of the writing. To address this, we will revise the manuscript to improve readability and flow. Specifically, we will make sure to highlight the potential extensions of this work in practice for the final version of the paper.
>
> **Rationale behind domain sensitivity:** The observed hidden state patterns suggest domain-specific understanding primarily through experimental evidence rather than a direct assertion. As outlined in the paper, we explored hidden states across multiple architectures and datasets to observe whether consistent patterns emerged. This approach is rooted in the idea that domain-specific understanding should manifest in the way a model activates and processes information across layers when presented with context-specific inputs [1][2][3]
>
> The patterns we observe are not inherently conclusive but show distinct trends that suggest domain sensitivity. For example, across various domains like medical, legal, and mathematical tasks, we saw that specific layers (second half of layers) tended to activate more strongly in response to relevant tokens or concepts tied to each domain. This observation was consistent across multiple model architectures, which strengthens the argument that these patterns are not simply artifacts of model design but rather indicative of domain-related reasoning processes.
>
> However, we acknowledge that these patterns do not prove a deep, inherent understanding of the domain in a traditional sense. Rather, they reflect domain-related processing and response patterns that differentiate between various types of information (e.g., factual, procedural, argumentative) across domains. The variation in these activations across datasets further supports the notion that the models are exhibiting domain-specific behavior, even if it remains somewhat superficial in terms of understanding. Therefore, our experimental methodology—testing across different architectures, finetuned models and using a wide variety of datasets—aims to provide evidence for domain sensitivity in hidden states, even as the exact nature of this sensitivity remains a subject of further investigation.
>
> **Further applications of this work:** The insights from analyzing domain-specific hidden state patterns can be applied in several practical ways:
>
> - _Steering Model Behavior:_ Modifying hidden states in real-time, towards “well-known” trajectories we can guide the model’s output toward specific behaviors or styles, such as using steering vectors to influence in specific domain generation process.
> - _Data/Training Efficiency:_ The hidden state traces can act as a proxy to identify which features drive faster convergence, enabling more efficient data use (e.g., the “Lottery Tickets Dataset”).
> - _Sparse Model Optimization:_ In our recent experiments on Phi-3.5-moe we have found that Sparse models exhibit predictable activation patterns in their hidden states of early layers. This insight could potentially allow for early prediction of which experts will be activated, enabling dynamic load balancing and smarter compute allocation.
>
> We leave these research angles for future directions as extensions. We thank again to the reviewer for their detailed feedback, and we will include the suggestions into the final version of the paper.
>
> [1] Can LLMs Infer Domain Knowledge from Code Exemplars? A Preliminary Study.
>
> [2] Linearity of relation decoding in transformer language models.
>
> [3] Inspecting and editing knowledge representations in language models.

---

### Official Review · Reviewer_kKJs · 2024-11-10

**Soundness:** 2
**Presentation:** 2
**Contribution:** 2
**Rating:** 3
**Confidence:** 4

**Summary:**

This paper makes the observation that the hidden states from autoregressive LLMs can separate data from different domains using the mean and variance of the activations. Using this property, a classifier is trained to predict the domain of the input from the hidden states and then route the example to a corresponding (domain specific) model.

**Strengths:**

- The observation that domains can be separated in the activation space by autoregressive LMs but not masked LMs (e.g., Deberta) is quite interesting. (However, I have an alternative explanation below that should be tested.)
- The idea of looking at the traces across layers is useful.

**Weaknesses:**

- Important details are missing regarding the methods and the experiments. （See questions below)
- The main claims need further evidence:
    - Figure 2 shows that Deberta doesn’t show a separation as other autoregressvie LLMs. But it’s also a much smaller model (86M vs >2B). It’s possible that such separation only shows in larger models. It’s good to test on an autoregressive LM of similar size such as GPT2.
    - The domain separation on MMLU is clear. However, when incorporating multiple datasets in Figure 2, it appears that math and medical domains (the green and blue lines) aren’t well separated (e.g., MedMCQA and GSM8k). Also, mean separation is not shown.
    - For the example routing results, almost all variation comes from the MATH and GSM8k dataest. More analysis would be helpful. E.g., the general performance on MATH is fairly low; how large is the dataset and what’s the standard error? What domains are predicted?
    - Also, I don’t quite understand the routing result. Is the idea to route each example to a different domain classifier? But why would examples from math datasets (say GSM8k) benefit from say a medical domain model?

Typo:
133: dim -> dimension

**Questions:**

- Equation 1: A is undefined. If A is the output of the normalization layers, shouldn’t the mean be always zero? Also, are the LLMs tested using the same type of normalization (or architecture)?
- Which LLM does the hidden states come from in Table 2?
- 379: why finetune on emotional data?
- 388: Are the Phi-3 models the finetuned models in step 1?
- 392: In step 4, what’s the input features to the MLP layer? Is it concatenated hidden states of all layers of the last token? But then, how do you change the number of layers; maybe it’s a sum of the hidden states of all layers?
- What is LLM sequence classifier?

---

> ### Author Response · Authors · 2024-11-22
> **Authors Response (Part 1)**
>
> We sincerely thank the reviewer for their insightful comments and constructive feedback.  Below, we address each point raised.
>
> **DeBERTa's lack of separation stems from model size:** We appreciate your observation regarding size differences influencing domain separation in DeBERTa. Additional experiments on autoregressive models of similar size (GPT-2, GPT-Neo, OPT at 125M parameters) confirm that separation patterns persist in smaller autoregressive models, though less pronounced. These findings will be included in the appendix, along with updates to Section 5.1 to clarify the model size’s role.
>
> **Overlap between medical and math traces:** We agree with your observation of a significant overlap across these domains. In the detailed analysis of our hypothesis in Appendix A.5, we explain that this overlap may stem from the shared structural reasoning processes required in these fields. In contrast, we observe a smaller overlap in the fields of laws and humanities, where the reasoning process relies more heavily on persuasive argumentation.
>
> **Routing results and dataset performance:** The primary goal of using hidden state activations for routing is to demonstrate their potential in selecting a model that best aligns with the unique characteristics of the input query. This is particularly valuable in unsupervised routing for open-ended and closed generative tasks, as the routing is guided by hidden state patterns rather than explicit sample labels.
>
> To train the router, we utilized hidden states from the Phi-3-mini-128k model applied to 4,000 random samples from the MMLU dataset (Base Pool). During inference, we evaluated the router on 1,000 unseen samples from each dataset listed in Table 2, including GSM8K, MATH, MEDMCQA, USMLE, and CASEHOLD. The router learned to classify queries into four domains—maths, biomedical, law, and humanities—and used this classification to route queries to the corresponding fine-tuned models for each domain.
> Additional clarifications:
> - the performance metrics in Table 2 are based on these 1,000 samples per dataset.
> - Standard errors and dataset sizes will be included in the final version.
>
> Regarding cases where, e.g., a mathematical query may benefit from a medical-domain model: in our experiments, we observed that the performance differences between domain-specific models can be small for certain tasks, suggesting that reasoning patterns or shared latent representations may overlap across domains. This overlap enables the router to identify subtle but meaningful connections that influence routing decisions effectively.
> Regarding the observed lower performance on the MATH and GSM8k datasets, this is attributable to the nature of the tasks. These datasets primarily contain open-ended, complex questions, which are inherently more challenging than constrained formats like multiple-choice questions. This reinforces the value of our approach in navigating these complex scenarios and highlights the potential of hidden state patterns in improving task-specific model selection.
>
> **Equation 1:** While layer normalization ensures a mean of zero for each token across its feature dimensions during the normalization step, the activations per layer (A) represent the final layer outputs, which include additional transformations (residual connections and learned bias terms applied after normalization); these transformations alter the mean, making it non-zero when aggregated across the batch and feature dimensions. Additionally, when aggregating across the batch and feature dimensions, variability in the inputs and context further contributes to deviations from zero.
>
> **Table 2:** The hidden states in Table 2 are derived from Base Pool samples obtained with Phi-3-mini-128k model (not finetuned) to maintain consistency with the traces across its finetuned counterparts. As discussed in Section 5.1 and Appendix A.4, we show that these traces persist across the different finetuned versions, ensuring consistent interpretability across different versions of the same model.
>
> **(L379-L388) Regarding fine-tuned models used in the router:** to clarify, the checkpoints from Hugging Face used in our experiments were not fine-tuned by us (due to resource constraints). Instead, we selected stable, publicly available checkpoints tailored to different domains, all based on the same small base model used in our experiments (Phi-3-mini-128k). Specifically, we selected high-quality models fine-tuned on math, medical, and emotional domains. By "high-quality” we mean checkpoints with a clear training process and strong performance on unseen datasets. These are the four models (3 finetuned + pretrained) evaluated in step 3 of section 5.3. After evaluating their performance, we identified the strongest checkpoints, as detailed in Appendix A.4.

---

> > ### Author Response · Authors · 2024-11-22
> > **Authors Response (Part 2)**
> >
> > **Regarding the input features to the MLP layer:** As mentioned in Section 5.1, we extracted the raw hidden state activations from each layer, focusing on the last token in the input query. This process is consistent across all activations analyzed in the paper. The resulting tensor has the shape (batch_size, dim, num_layers), where num_layers corresponds to the number of layers in the model. The MLP directly takes this tensor as input, learning to discriminate from these activations without requiring concatenation or summation across layers. While we acknowledge that an MLP may not be the optimal architecture for this task, we utilized it as a proof of concept to demonstrate the potential of extracting meaningful patterns from hidden state activations across layers. We provide a careful description of the MLP layer in Appendix A.2.
> >
> > **Regarding the LLM sequence classifier:** The LLM Sequence Classifier leverages the Phi-3-mini-128k model in a zero-shot classification framework. In this setup, the task involves providing the model with an input context and executing multiple forward passes to evaluate all available options. The option with the highest log-likelihood is then selected as the predicted answer.
> >
> > This approach contrasts with our use of hidden states for routing tasks. Hidden states are extracted in a single forward pass during the prefill phase, which significantly reduces computational overhead. By relying solely on hidden states, we avoid the need for iterative evaluations across multiple outputs, making the process more efficient for real-time or resource-constrained applications.
> >
> > We again thank the reviewer for their thoughtful comments, and we will incorporate this feedback into the final version of our paper.

---

### Meta-Review · Area_Chair_gF2f · 2024-12-23

**Metareview:**

Reject by reviewer consensus.

This paper makes the observation that the hidden states from autoregressive LLMs can separate data from different domains using the mean and variance of the activations. Using this property, a classifier is trained to predict the domain of the input from the hidden states and then route the example to a corresponding (domain specific) model, leading to improved performance in cross-domain generalization tasks like legal, medical, and mathematical reasoning.

Reviewers generally liked the direction and found the claims of the paper to be clear. However, some reviewer are not convinced of the conclusions (kKJs: large vs. small size instead of model type; uiqH: applicability beyond the chosen domains, 2yn5: questioning experimental setups).

**Additional Comments On Reviewer Discussion:**

authors responded fairly late in the cycle, no reviewer response, not too surprising since it's all rejects.

---

### Decision · Program_Chairs · 2025-01-22

Reject